# Clinical and Tumor Characteristics of Patients with High Serum Levels of Growth Differentiation Factor 15 in Advanced Pancreatic Cancer

**DOI:** 10.3390/cancers13194842

**Published:** 2021-09-28

**Authors:** Hidetaka Suzuki, Shuichi Mitsunaga, Masafumi Ikeda, Takao Aoyama, Kazumi Yoshizawa, Hiroki Yoshimatsu, Norisuke Kawai, Mari Masuda, Tomofumi Miura, Atsushi Ochiai

**Affiliations:** 1Division of Biomarker Discovery, Exploratory Oncology Research & Clinical Trial Center, National Cancer Center, Kashiwa 277-8577, Japan; hidesuzu@east.ncc.go.jp (H.S.); tomiura@east.ncc.go.jp (T.M.); aochiai@east.ncc.go.jp (A.O.); 2Laboratory of Pharmacotherapeutics, Faculty of Pharmaceutical Science, Tokyo University of Science, Tokyo 278-8510, Japan; t-aoyama@rs.noda.tus.ac.jp; 3Department of Pharmacy, National Cancer Center Hospital East, Kashiwa 277-8577, Japan; 4Department of Hepatobiliary and Pancreatic Oncology, National Cancer Center Hospital East, Kashiwa 277-8577, Japan; masikeda@east.ncc.go.jp; 5Laboratory of Pharmacology and Therapeutics, Faculty of Pharmaceutical Science, Tokyo University of Science, Tokyo 278-8510, Japan; yoshizawa-ph@rs.tus.ac.jp; 6Pfizer R&D Japan G.K., Tokyo 151-8589, Japan; Hiroki.Yoshimatsu@pfizer.com (H.Y.); Norisuke.Kawai@pfizer.com (N.K.); 7Department of Proteomics, National Cancer Center Research Institute, Tokyo 104-0045, Japan; mamasuda@ncc.go.jp

**Keywords:** growth differentiation factor 15, pancreatic cancer, anorexia, cancer cachexia

## Abstract

**Simple Summary:**

Growth differentiation factor 15 (GDF-15) is a stress responsive cytokine that mediates food intake, energy consumption, and body weight. We aimed to evaluate whether circulating GDF-15 level could be associated with cachexia symptoms, which include loss of skeletal muscle mass, systemic inflammatory reaction, poor performance status, anorexia, shortened survival time and biological tumor activity in advanced pancreatic cancer (APC). The cut-off for serum GDF-15 was 3356.6 pg/mL, as the mean plus two standard deviations in patients with benign pancreatic disease. APC patients with high serum GDF-15 showed worsened performance, anorexia and elevations of inflammatory and tumor burden, signatures of cachexia, and activation of Akt and JNK in tumor GDF-15-producing pathways. This study identified tumor-driven GDF-15 as a potential cause of cachexia symptoms in APC.

**Abstract:**

We aimed to evaluate the association of circulating growth differentiation factor 15 (GDF-15) with cachexia symptoms and the biological activity of advanced pancreatic cancer (APC). Treatment-naïve patients with liver metastasis of APC or with benign pancreatic disease were retrospectively analyzed. Clinical data, blood samples, and biopsy specimens of liver metastasis were collected prior to anti-cancer treatment. Serum GDF-15 levels and multiple protein expressions in lysates extracted from liver metastasis were measured by enzyme-linked immuno-sorbent assay and reverse-phase protein array, respectively. The cut-off for serum GDF-15 was determined as 3356.6 pg/mL, the mean plus two standard deviations for benign pancreatic disease. The high-GDF-15 group was characterized as showing low Karnofsky performance status (KPS) (*p* = 0.037), poor Eastern Cooperative Oncology Group performance status (ECOG-PS) (*p* = 0.049), severe appetite loss (*p* = 0.011), and high serum levels of carbohydrate antigen 19-9 (*p* = 0.019) and C-reactive protein (*p* = 0.009). Tumors of the high-GDF-15 group expressed high levels of phosphorylated (p)JNK (*p* = 0.007) and pAkt (*p* = 0.040). APC patients with high serum GDF-15 showed signatures of cachexia and activation of the signaling pathways involving Akt and JNK in the tumor. This study indicated circulating GDF-15 could be associated with cachectic symptoms in APC.

## 1. Introduction

As a member of the transforming growth factor (TGF)-β subfamily, growth differentiation factor 15 (GDF-15) is a stress-responsive cytokine associated with various diseases such as cancers [1,2,3,4,5], cardiovascular disorders [6], mitochondrial disorders [7], hyperthyroidism [8], obesity and type 2 diabetes [9]. GDF-15 mediates food intake, energy consumption, and body weight [10,11]. Circulating GDF-15 binds to the GDNF family receptor a-like (GRFAL)/RET receptor complex in the brain stem, triggering receptor phosphorylation and downstream intracellular signaling [10]. The signaling of GFRAL antagonism and agonism, respectively, increase and decrease food intake [10]. High circulating levels of GDF-15 in cancer patients have been associated with chemoresistance, anorexia, emesis and weight loss [12,13,14], all of which are symptoms of cachexia [15]. Cachexia is defined as a multiple metabolic disorder characterized by an ongoing loss of skeletal muscle mass [16]. This pathology is frequently seen in patients with advanced pancreatic cancer (APC) [17], who show systemic inflammation and worsened performance status (PS) [18,19,20]. One possibility is that elevated levels of circulating GDF-15 are associated with cachectic symptoms, including loss of skeletal muscle mass, systemic inflammatory reaction, poor PS, anorexia, and shortened survival time.

Expression of GDF-15 is maintained at low levels in most tissues [21] and is increased by p53 activation due to various cellular stressors, such as inflammation, oxidative stress, hypoxia, and oncogene activation [22]. Serum GDF-15 levels in the patients with chronic pancreatitis were higher, 2248 pg/mL in mean [1] compared to healthy subjects (416.8 pg/mL [23], 546 [24], 639 [25] in mean). Pancreatic disease compressing the pancreatic duct is considered to increase the GDF-15 level, and thus can result from benign pancreatic diseases including chronic pancreatitis, intraductal papillary mucinous tumor, and serous cystic tumor. The cut-off level to characterize a high serum level of GDF-15 in APC thus should discriminate cancer-related upregulation of GDF-15 from the effects of benign pancreatic disease. We therefore planned to determine the upper limit of serum GDF-15 levels in benign pancreatic diseases, in order to set an appropriate cut-off to discriminate low and high serum GDF-15 groups for APC.

Solid stress increased GDF-15 expression by transcriptionally regulating GDF-15 expression through activation of the Akt pathway in pancreatic cancer cells [26]. Stress responses involving the Akt pathway can become dominant in cases with pancreatic cancer complicated by high levels of circulating GDF-15, providing biological validation for a cut-off for serum GDF-15 in pancreatic cancer. The clinical and tumoral characteristics of APC patients with high serum GDF-15 levels are expected to provide useful information for elucidating the causes of clinical symptoms triggered by stress responses in the tumor microenvironment.

We undertook the present study to investigate the following in APC patients: (A) an appropriate cut-off value for serum GDF-15 levels; (B) the relationship between serum GDF-15 levels and clinical data; and (C) tumor characteristics associated with high serum levels of GDF-15.

## 2. Materials and Methods

### 2.1. Ethics

This study was approved by the ethics review committee of National Cancer Center Hospital East (approval no. 2020-373).

### 2.2. Patients

Treatment-naïve patients with liver metastasis from pathologically proven APC or with benign pancreatic disease were eligible to participate in this study. Subjects were registered between 5 August 2011 and 7 January 2015 at the National Cancer Center Hospital East. Benign pancreatic diseases comprised chronic pancreatitis, intraductal papillary mucinous tumor, non-invasive intraductal papillary mucinous carcinoma, and serous cystic tumor. Written informed consent was obtained from all subjects before enrollment.

### 2.3. Definition of Cancer Cachexia and Sarcopenia

In this study, we defined cancer cachexia as either weight loss >5% or weight loss >2% with a body mass index (BMI) < 20 kg/m^2^ [16]. Base cachexia was defined as weight loss within 6 months before starting chemotherapy. Sarcopenia was diagnosed using the Japan Society of Hepatology criteria [27] as low handgrip strength (<26 kg for males; <18 kg for females) and low muscle mass (<42 cm^2^/m^2^ for males; <38 cm^2^/m^2^ for females). Muscle mass was evaluated as the skeletal muscle index (SMI) at the level of the third lumbar vertebra (L3) on computed tomography.

### 2.4. Data Collection

Body weight, Karnofsky performance status (KPS), Eastern Cooperative Oncology Group performance status (ECOG-PS), laboratory tests, severity of symptoms (the Japanese version of the MD Anderson Symptom Inventory; MDASI-J) [28] and a blood sample were collected before anti-cancer treatment. MDASI-J is a 13-item symptom scale (pain, fatigue, nausea, disturbed sleep, distress, shortness of breath, memory, lack of appetite, drowsiness, dry mouth, sadness, vomiting, and numbness or tingling) rated numerically from 0 (“not present”) to 10 (“as bad as you can imagine”) based on symptoms present over the preceding 24 h. Severe appetite loss was defined as MDASI-J ≥ 2, as per our previous study [29]. The cut-off for serum levels of C-reactive protein (CRP) was determined as 2.0 mg/dL in accordance with our previous study [20]. The cut-off for carbohydrate antigen (CA)19-9 was defined as above median serum CA19-9 level, which was 4150 U/mL in our study.

### 2.5. Enzyme-Linked Immuno-Sorbent Assay (ELISA)

Blood samples were collected in a serum-separating tube on the morning after overnight fasting and completely clotted for 30–60 min at room temperature. After centrifugation at 1500 g, 25 °C for 15 min, serum samples were separated and frozen at −80 °C until analysis. Serum GDF-15 levels were quantitatively determined in duplicate using an ELISA kit in accordance with the protocol from the manufacturer (R&D Systems, Minneapolis, MN, USA).

### 2.6. Protein Expression Analysis

Tissue cores were obtained from liver metastases of APC with endoscopic ultrasonography-guided fine-needle aspiration biopsy (FNAB) [30]. Immediately after the procedure, a tissue core specimen was homogenized with tissue-Lyser II (QIAGEN, Hilden, Germany) in the lysis buffer of a NucleoSpin TriPrep kit (Macherey-Nagel, GmbH & Co., KG, Düren, Germany), and was filtered with the NucleoSpin TriPrep kit. After protein purification with a Pierce™ SDS-PAGE Sample Prep Kit (Thermo Fisher Scientific, Waltham, MA, USA), the lysate was snap-frozen and stored at −80 °C until needed. Reverse-phase protein array (RPPA) analysis of lysates was performed at the functional proteomic RPPA core facility of the MD Anderson Cancer Center. Briefly, tissue lysates were serially diluted two-fold for five dilutions (Undiluted, 1:2, 1:4, 1:8, and 1:16) and arrayed onto nitrocellulose-coated slides [31,32]. RPPA slides immunostained with 435 primary antibodies were scanned with a TissueScope scanner (HURON Digital Pathology, St. Jacobs, ON, Canada), and the signal intensity of each spot on the arrays was quantified using Array-pro Analyzer 6.3 software (Media Cybernetics, Silver Spring, MD, USA). The relative protein levels of analytes in each sample were then determined using the SuperCurve 1.5.0 R package via SuperCurveGUI 2.1.1. [33]. RPPA analysis was performed in the patients whose protein samples could be extracted, purified, and detected as a reproducible band of β-actin in western blotting.

### 2.7. Statistical Analyses

Continuous data were compared using the Mann–Whitney *U* test. Associations between serum GDF-15 levels and each categorical variable were examined using a chi-square test or Fisher’s exact test. Fisher’s exact test and chi-square test were used for analysis of 2 × 2 contingency tables and more than 2 × 2 consistency tables, respectively. Overall survival (OS) was calculated from the starting date of first-line chemotherapy or BSC. Progression-free survival (PFS) was calculated only in patients with chemotherapy from the starting date of first-line chemotherapy. Survival curves were drawn using the Kaplan–Meier method. OS and PFS were compared using the log-rank test between low- and high-GDF-15 groups of APC patients with liver metastasis. For all statistical tests, values of *p* < 0.05 were considered statistically significant. Median survival times were calculated with 95%CIs, as determined using the Brookmeyer and Crowley method. Data analyses were performed using JMP^®^ version 11 software (SAS Institute, Cary, NC, USA).

## 3. Results

### 3.1. Cut-Off Value for Serum GDF-15 Levels

The CONSORT diagram for this study is shown in Figure 1. A total of 120 patients were included and analyzed in this study, of whom 99 patients had liver metastasis of APC (38 women, 61 men; median age, 66.2 years; range, 39–85 years). The remaining 21 patients had benign pancreatic disease (10 women, 11 men; median age, 69.0 years; range, 50–81 years), including chronic pancreatitis (*n* = 4, 19.1%), intraductal papillary mucinous tumor (*n* = 15, 71.4%) and serous cystic tumor (*n* = 2, 9.5%).

Mean (± standard deviation [SD]) serum GDF-15 levels were significantly higher in the 99 patients with liver metastasis of APC (2990.2 ± 1967.0 pg/mL) than in the 21 patients with benign disease (1177.0 ± 1089.8 pg/mL; *p* < 0.001) (Figure 2). The cut-off value was determined as 3356.6 pg/mL, representing the mean plus twice the SD of the serum GDF-15 level in patients with benign disease, (Figure 2). As a result, 34.3% (*n* = 34) of the 99 APC patients were classified as the high-GDF-15 group. Mean serum GDF-15 levels were 5257.0 ± 1649.9 pg/mL in the high-GDF-15 group and 1804.5 ± 602.5 pg/mL in the low-GDF-15 group.

### 3.2. Baseline Clinical Characteristics

Among the 99 APC patients, 87 patients (87.9%) received first-line chemotherapy, comprising gemcitabine + nab-paclitaxel (*n* = 3, 3.0%); FOLFIRINOX (fluorouracil, leucovorin, irinotecan, and oxaliplatin) (*n* = 14, 14.1%); gemcitabine (*n* = 26, 26.3%); gemcitabine + erlotinib (*n* = 32, 32.3%); and others (*n* = 12, 12.1%). The remaining 12 patients (22.1%) received the best supportive care (BSC). The characteristics of patients with high serum GDF-15 levels included low KPS (*p* = 0.037), poor ECOG-PS (*p* = 0.049), high serum CA19-9 levels (*p* = 0.019), high serum CRP levels (*p* = 0.009), and severe appetite loss (*p* = 0.011) (Table 1). The proportion of patients receiving chemotherapy was significantly lower in the high-GDF-15 group (76.5%) than in the low-GDF-15 group (93.9%, *p* = 0.020) (Table 1). No significant differences in other baseline clinical data were apparent between high- and low-serum GDF-15 levels, including proportion of chemotherapy regimen.

Actual mean serum GDF-15 levels were high in groups with severe appetite loss (*n* = 48), high CRP (*n* = 43), KPS < 90 (*n* = 42), and CA19-9 > 4150 U/mL (*n* = 49) as compared to groups with non-severe appetite loss (*n* = 43; 3407.1 pg/mL vs. 2285.9 pg/mL; *p* = 0.003; Figure 3A), low serum CRP (*n* = 56; 3581.0 pg/mL vs. 2535.5 pg/mL; *p* = 0.013; Figure 3B), KPS ≥ 90 (*n* = 49; 3274.32 pg/mL vs. 2367.3 pg/mL; *p* = 0.001; Figure 3C), and CA19-9 ≤ 4150 U/mL (*n* = 50; 3348.3 pg/mL vs. 2639.2 pg/mL; *p* = 0.027; Figure 3D), respectively.

### 3.3. Survival

Median OS was 4.10 months (95% confidence interval [95%CI], 3.33–5.87 months) for all 99 patients with APC, 2.70 months (*n* = 34; 95%CI, 1.37–5.83 months) for high-GDF-15 patients and 5.70 months (*n* = 65; 95%CI, 3.63–7.60 months) for low-GDF-15 patients (*p* = 0.113).

In the 87 patients who received chemotherapy, the median OS was 5.57 months (95%CI, 3.63–6.53 months). Median OS was 3.33 months (*n* = 26; 95%CI, 1.73–5.87 months) for high-GDF-15 patients and 5.87 months (*n* = 61; 95%CI, 3.70–7.97 months) for low-GDF-15 patients. No significant differences in OS were evident between high- and low-GDF-15 patients (*p* = 0.254) (Figure 4A).

Median PFS was 2.20 months (95%CI, 1.83–2.87 months) for the 87 patients with chemotherapy, 1.58 months (*n* = 26; 95%CI, 0.97–2.73 months) for high-GDF-15 patients, and 2.60 months (*n* = 61; 95%CI, 1.90–3.20 months) for low GDF-15 patients (*p* = 0.546). No significant differences in PFS were apparent between high- and low-GDF-15 patients (Figure 4B).

### 3.4. Tumor Characteristics

RPPA analysis was performed in 38.2% (*n* = 13) of high-GDF-15 patients (*n* = 34) and 41.5% (*n* = 27) of low-GDF-15 patients (*n* = 65). In this study, 435 proteins and phosphoproteins were analyzed in the lysates of FNAB tissue cores from liver metastases of APC patients. Significant differences between the low- and high-GDF-15 groups were found for 12 proteins (Appendix A), including phospho-(p)JNK (threonine183_tyrosine185) and pAkt (threonine 308), both of which were significantly higher in the high-GDF-15 group than in the low-GDF-15 group (pJNK: 0.046 vs. −0.094 in mean, *p* = 0.007, Figure 5A; pAkt: 0.123 vs. −0.020, *p* = 0.040, Figure 5B).

The 40 APC patients who underwent tumor analysis by RPPA were separated into two groups according to median expression levels (pJNK: −0.0256 and pAkt: 0.0036). Mean serum GDF-15 levels were significantly higher in the groups with pJNK expression level ≥ −0.0256 (*n* = 19) and pAkt expression level ≥ 0.0036 (*n* = 21) than in the groups with pJNK expression level < −0.0256 (*n* = 21; 3522.6 pg/mL vs. 2040.5 pg/mL; *p* = 0.015; Figure 5C) and pAkt expression level < 0.0036 (*n* = 19; 3307.8 pg/mL vs. 2122.0 pg/mL; *p* = 0.020; Figure 5D).

## 4. Discussion

Mean circulating GDF-15 levels have been suggested to be higher in patients with pancreatic ductal adenocarcinoma, as 1731 pg/mL [23], 2428 [24], 7694.58 [1] and 2990.2 (this study), compared to patients with colorectal adenocarcinoma (CRC) (1371.0 pg/mL [23] and 2030 pg/mL [25]), non-small-cell lung carcinoma (mean, 1258.0 pg/mL [23] and median, 828.8–2000 pg/mL [34,35]), and gastric adenocarcinoma (1154.0 pg/mL [23]). An atlas of the human transcriptome reported that GDF-15 mRNA expression was relatively low in pancreatic cancer compared to prostate, urothelial, renal, melanoma, and colorectal cancers [36]. The discrepancy between tumor mRNA expression and circulating level of GDF-15 may involve the ratio of patients with metastasis in each study population. A large proportion of CRC patients in a previous study [23] showed early-stage disease, although CRC patients with distant metastasis showed higher levels of serum GDF-15 (mean, 3531 pg/mL) [25] and all APC patients in the present study showed liver metastasis. GDF-15 levels in various cancers should thus be assessed in concert with the ratio of metastasis in the study population. The diurnal pattern of serum GDF-15 levels needs to be clarified for evaluation of serum GDF-15. Circadian oscillation led to variations in serum GDF-15 levels near the midline estimating statistic of rhythm (MESOR) of about ± 10%, with a MESOR of 381 ± 15 pg/mL in healthy men [37]. In the present study, blood samples were obtained the morning after overnight fasting, and all APC patients showed liver metastasis. The effects of circadian oscillation and metastatic rate on circulating GDF-15 levels were considered to have been minimized in our study.

Our results showed that patients in the high-GDF-15 group experienced severe appetite loss compared with those in the low-GDF-15 group. In a previous study, median serum GDF-15 levels were higher in cancer patients with anorexia (1224.1 pg/mL) than in patients without anorexia (812.4 pg/mL) [34], where the cut-off value for anorexia was set at the lowest Functional Assessment of Anorexia/Cachexia Treatment (FAACT) score tertile, similar to the consensus definition of anorexia (FAACT ≤ 24) [38]. Our results showed that median MDASI scores in the high- and low-GDF-15 groups were 4 and 1 on a 0–10 scale, respectively. National Comprehensive Cancer Network (NCCN) guidelines for palliative care have proposed defining non-pain symptoms scoring >4 on the 0–10 scale as severe [39]. In validation studies of MDASI, severe, moderate, and mild symptoms are defined as scores ≥7, 5 or 6, and 1–4, respectively [28,40]. The degrees of anorexia in high- and low-GDF-15 groups corresponded to the upper and lower limits of mild anorexia, respectively. A prior report proposed the minimal clinical difference in numerical rating scale for anorexia as 1.5 points [41], and the median MDASI-J score in our study was 3 points higher in the high-GDF-15 group than in the low-GDF-15 group. On this basis, we concluded that patients in the high-GDF-15 group showed a clinically meaningful increase in the severity of anorexia compared with those in the low-GDF-15 group.

Our RPPA analysis revealed that expressions of pJNK and pAKT were significantly higher in the high-GDF-15 group than in the low-GDF-15 group. Tumor-associated stresses such as solid stress have been reported to induce phosphorylation of JNK and AKT [42,43,44]. Growth-induced solid stress in a tumor reflects an excessive tumor burden that compresses surrounding tissues [45,46]. APC patients with high serum GDF-15 levels also had high serum CA19-9 levels, suggesting high tumor burden [47]. Intriguingly, Akt and JNK have been shown to activate transcription of the GDF-15 gene [26,48] and to promote secretion of the GDF-15 protein [5,49]. We therefore speculate that increased solid stress due to a large tumor burden augments the expressions of pJNK and pAKT and also increases levels of circulating GDF-15. In addition, we observed that p-forkhead box-containing protein (FOX)O3a (serine 318_serine 321), p-tuberin (threonine_1462), and pElk-1 (serine 383) were highly expressed in the high-GDF-15 group compared to the low-GDF-15 group (Appendix A). Activation of FOXO3a and tuberin was noted to promote tumor growth [50,51], which was inhibited by Akt-mediated phosphorylation [52]. On the other hand, Elk1 was activated by JNK-mediated phosphorylation, leading to cell proliferation and oncogenesis [53]. These observations suggest that signaling pathways involving Akt and JNK play important roles in tumor growth in APC patients with high serum GDF-15 levels.

Evans’ diagnostic criteria for cachexia include body weight loss, sarcopenia, systemic inflammation, appetite loss and fatigue, and are related to poor PS [54]. In our study, patients with high serum levels of GDF-15 showed severe appetite loss, systemic inflammation and poor PS, but not sarcopenia or body weight loss. We consider that an increase in tumor burden contributed to GDF-15-related anorexia via solid stress and was related to systemic inflammation and worsened PS prior to the onset of body weight loss and sarcopenia. A phase 1, first-in-humans, two-part, open-label clinical trial was recently started for CTL-002, a GDF-15 neutralizing antibody, recruiting subjects with advanced-stage, relapsed, or refractory solid tumors (NCT04725474). The target patients will be evaluated for not only antitumor effect and safety, but also cachexia-related outcomes, since inhibition of GDF-15 biological activity has been shown to reverse body weight loss and restore muscle and fat tissue mass in several cachectic animal models [55]. This clinical trial of CTL-002 is expected to clarify the relationship between GDF-15 and cachexia.

The present study had a limitation that requires consideration when interpreting the results. This study had no validation cohort for the cut-off value of serum GDF-15 levels. Our cut-off value was able to discriminate the biological activity of JNL and AKT signaling in tumors as pathways producing GDF-15. In terms of biological activity, our cut-off value for serum GDF-15 was confirmed. Further studies to validate our cut-off for serum GDF-15 are warranted.

## 5. Conclusions

In conclusion, APC patients with high serum GDF-15 showed worsened performance status, anorexia and elevations of inflammation and tumor burden as signatures of cachexia, as well as activation of Akt and JNK in tumor GDF-15-producing pathways. This study indicated that circulating GDF-15 could be associated with cachectic symptoms in APC.

## Figures and Tables

**Figure 1 cancers-13-04842-f001:**
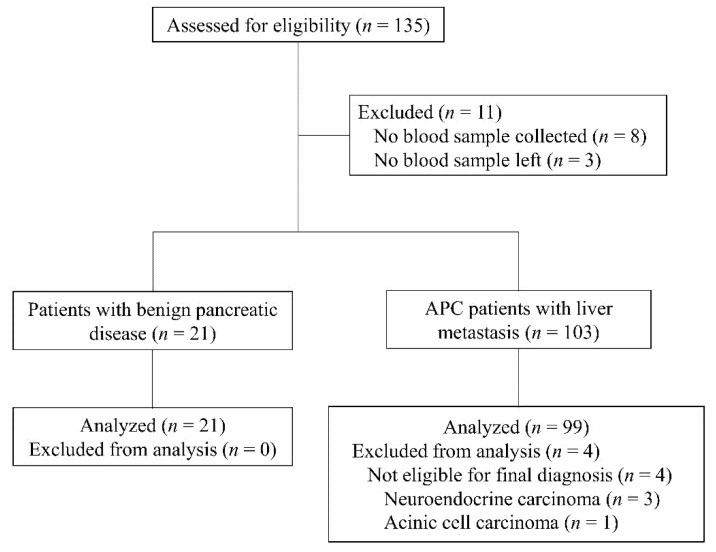
Consort diagram. APC, advanced pancreatic cancer.

**Figure 2 cancers-13-04842-f002:**
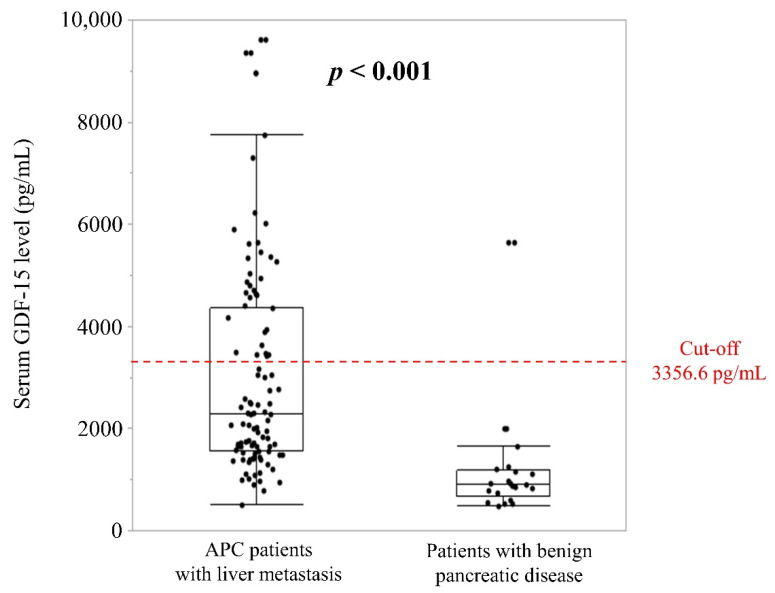
Serum GDF-15 levels in APC patients with liver metastasis and benign pancreatic diseases. The red dotted line represents the cut-off for serum GDF-15 levels, determined as the mean plus twice the standard deviation of GDF-15 in patients with benign pancreatic disease, equal to 3356.6 pg/mL.

**Figure 3 cancers-13-04842-f003:**
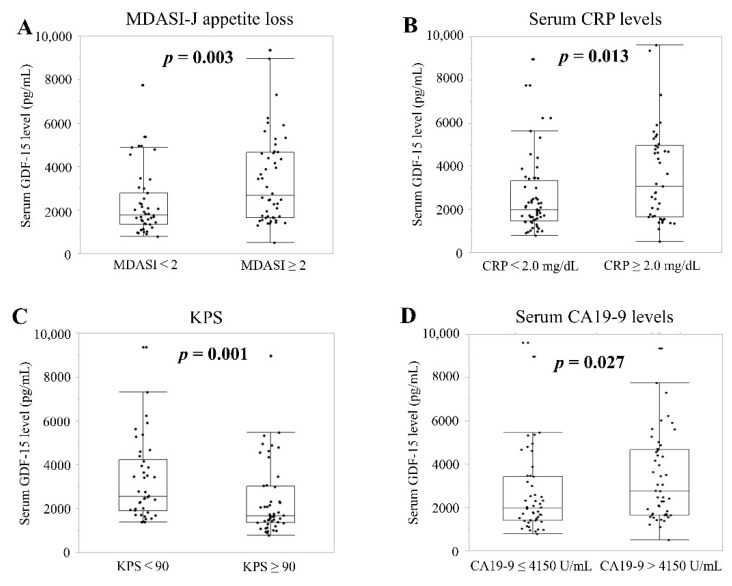
Relationship between serum GDF-15 levels and clinical parameters related to cachexia and disease progression in patients with liver metastasis of APC. Parameters include (**A**) appetite loss; (**B**) CRP; (**C**) KPS; and (**D**) CA19-9. MDASI-J, Japanese version of the MD Anderson Symptom Inventory; CRP, C-reactive protein; KPS, Karnofsky performance status; CA19-9, carbohydrate antigen 19-9.

**Figure 4 cancers-13-04842-f004:**
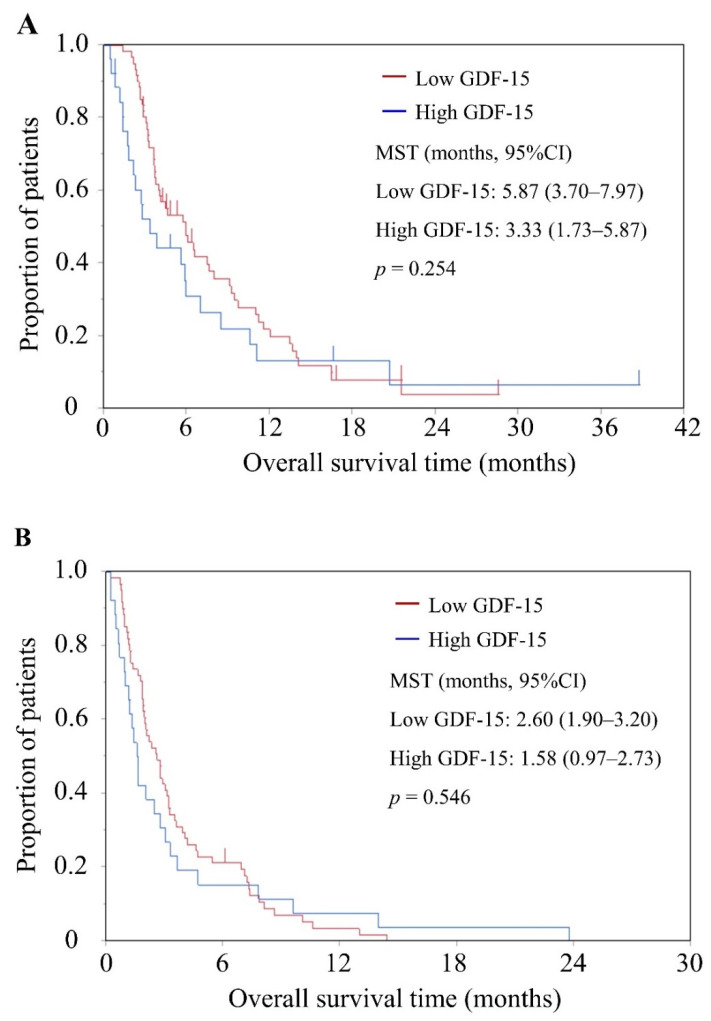
Overall survival (OS) and progression-free survival (PFS) according to high or low serum GDF-15 levels in patients with liver metastasis of APC. Kaplan-Meier plots of (**A**) OS and (**B**) PFS for patients receiving chemotherapy. MST, median survival time.

**Figure 5 cancers-13-04842-f005:**
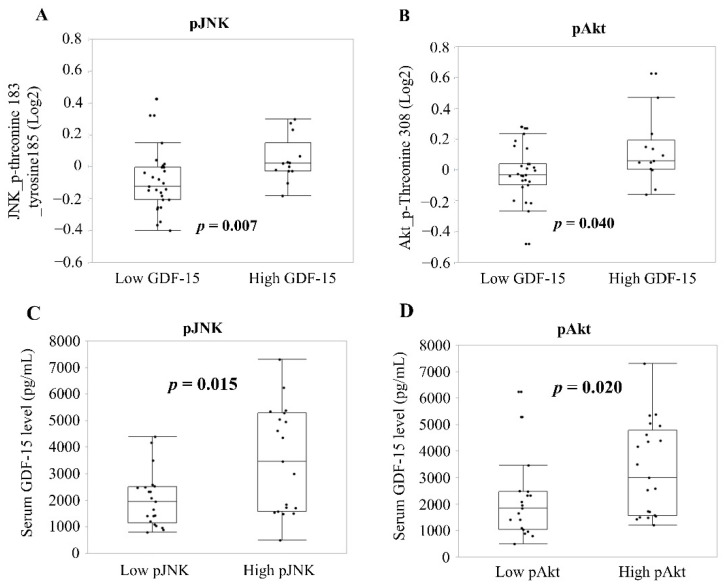
Tumor protein expression associated with serum GDF-15 levels by RPPA. Tumor protein expressions of (**A**) pJNK and (**B**) pAkt, represented as a binary logarithm, were compared among patients in the low- and high-GDF-15 groups. Serum GDF-15 levels, represented as logGDF-15, were compared among patients with high and low protein expressions of (**C**) pJNK and (**D**) pAkt in the tumor. Boxes indicate 5th and 95th percentiles. pJNK, phospho-JNK; pAkt, phospho-Akt.

**Table 1 cancers-13-04842-t001:** Univariate analysis of baseline clinical characteristics to identify factors related to serum GDF-15 levels in patients with metastatic APC.

Variables		Serum GDF-15 Levels		*p*-Value
		Low	High	
Sex	Male	42 (64.6%)	19 (55.9%)	0.514
	Female	23 (35.4%)	15 (44.1%)	
Age (years)	Median (range)	67.0 (39–85)	65.9 (50–85)	0.665
KPS	Median (range)	90 (60–100)	80 (60–100)	0.037
ECOG-PS	0	36 (56.3%)	9 (31.0%)	0.049
	1	25 (39.1%)	16 (55.2%)	
	2	3 (4.7%)	4 (13.8%)	
Chemotherapy	Yes	61 (93.9%)	26 (76.5%)	0.020
Regimen	GEM + nab-PTX	2 (3.3%)	1 (3.8%)	0.738
	Modified FOLFIRINOX	10 (16.4%)	4 (15.4%)	
	GEM monotherapy	16 (26.2%)	10 (38.5%)	
	GEM + Erlotinib	23 (37.7%)	9 (34.6%)	
	others	10 (16.4%)	2 (7.7%)	
CEA (ng/mL)	Median (range)	14.8 (0.9–363.4)	26.4 (1.3–1441.0)	0.247
CA19-9 (U/mL)	Median (range)	2995.0 (0.1–330,600.0)	10771.5 (0.5–343,352.0)	0.019
Ascites	Present	34 (52.3%)	14 (41.2%)	0.397
Peritoneal metastasis	Present	28 (43.1%)	19 (55.9%)	0.290
Biliary Drainage before starting chemotherapy	Present	8 (12.3%)	5 (14.7%)	0.760
Lung metastasis	Present	14 (21.5%)	6 (17.7%)	0.794
Distant lymph node metastasis	Present	4 (6.1%)	5 (14.7%)	0.268
Serum CRP levels (mg/dL)	Median (range)	1.20 (0.03–18.58)	3.22 (0.06–21.72)	0.009
Cachexia	Cachexia	37 (56.9%)	17 (50.0%)	0.531
	Non-cachexia	28 (43.1%)	17 (50.0%)	
Sarcopenia	Sarcopenia	19 (33.9%)	10 (52.6%)	0.179
	Non-sarcopenia	37 (66.1%)	9 (47.4%)	
SMI (kg/m^2^)	Median (range)	44.26 (27.40–52.10)	40.14 (28.28–52.88)	0.203
MDASI-J appetite loss	Median (range)	1 (0–10)	4 (0–10)	0.011
Body weight loss (%)	Median (range)	5.74 (−17.42–22.81)	5.03 (−1.72–30.00)	0.690

Fisher’s exact test or the chi-square test were used for comparisons of categorical variables between the low-GDF-15 group (serum GDF-15 level < 3356.6 pg/mL) and the high-GDF-15 group (serum GDF-15 level ≥ 3356.6 pg/mL). KPS, Karnofsky performance status; ECOG-PS, Eastern Cooperative Oncology Group performance status; GEM, gemcitabine; nab-PTX, nab-paclitaxlel; CEA, carcinoembryonic antigen; CA19-9, carbohydrate antigen 19-9; CRP, C-reactive protein; SMI, skeletal muscle mass index; MDASI-J, Japanese version of the MD Anderson Symptom Inventory.

## Data Availability

The datasets used and/or analyzed during the present study are available from the corresponding author on reasonable request.

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
