# Peer review of "Clinical and Tumor Characteristics of Patients with High Serum Levels of Growth Differentiation Factor 15 in Advanced Pancreatic Cancer"

_cancers, 2021, doi:10.3390/cancers13194842_

Round 1

Reviewer 1 Report

The authors present an interesting study. They aim at establishing a cut-off level for serum GDF levels in patients with advanced pancreatic cancer patients.

Comment 1: The authors compare the serum GDF levels in patients with benign pancreatic disorders and advanced pancreatic cancers. It would be interesting to see the levels of serum GDF in normal, healthy people/donors.

Comment 2: Line 328: “This study indicated tumor-driven GDF-15 as a potential cause of cachexia symptoms in APC.”

This needs to be re-written/clarified.

Based on lines 180-182 and 194-197, it seems that GDF15 levels and appetite loss are correlated. There is no direct evidence that GDF-15 is a driver/cause of cachexia. These the message that in patients with high serum GDF appetite loss and other symptoms were observed and vice versa- in patients with more sever appetite loss, there was high serum GDF.

Lines 180-182: “The characteristics of patients with high serum GDF- 180 15 levels included low KPS (P = 0.037), poor ECOG-PS (P = 0.049), high serum CA19-9 181 levels (P = 0.019), high serum CRP levels (P = 0.009), and severe appetite loss (P = 0.011) 182 (Table 1).”

Lines 194-197: “Actual mean serum GDF-15 levels were high in...”

Comment 3: Line 228 :“RPPA analysis was performed in 38.2% (n = 13) of high-GDF-15 patients (n = 34) and 228 41.5% (n = 27) of low-GDF-15 patients (n = 65)”

How were these patients selected: was it randomized or was there a selection criterion?

Comment 4: Given that there is no difference in patient survival in the cohorts with low and high GDF-15 levels, it would be necessary to emphasize the importance/clinical relevance of circulating GDF-15. There is no direct evidence in this manuscript that high GDF-15 causes cachexia.

Reviewer 2 Report

The study is well designed and indicates how tumor-driven GDF-15 is a potential cause of cachexia in advanced pancreatic cancer.

Author Response

We would like to thank you for taking the time and effort necessary to review the manuscript.

Reviewer 3 Report

The manuscript demonstrates that growth differentiation factor-15 (GDF15) is a reliable biomarker for prognostication of advance pancreatic cancer (APC). The authors further demonstrated that APC patients with high serum GDF15 levels showed poor survival, anorexia and high levels of inflammation and tumor burden. This effect correlated with activation of Akt and JNK in the tumor. Overall, the work is interesting and aligns with the previous publication where GDF-15 has been proposed to be more reliable APC biomarker and independent predictor of APC diagnosis than CA19.9. There are few suggestions.

  1. Some additional information about GDF-15 in the introduction section is needed.
  2. The authors should discuss and compare results from previous studies on GDF-15 and CA19.9 and other prognostic markers in APC.
